# Management of Neonates in the Special Care Nursery and Its Impact on the Developing Gut Microbiota: A Comprehensive Clinical Review

**DOI:** 10.3390/microorganisms13081772

**Published:** 2025-07-29

**Authors:** Ravisha Srinivasjois, Shripada Rao, Gavin Pereira

**Affiliations:** 1Department of Neonatology and Paediatrics, Joondalup Health Campus, Perth, WA 6027, Australia; 2Medical School, University of Western Australia, Perth, WA 6005, Australia; shripada.rao@health.wa.gov.au; 3Department of Health Sciences, Edith Cowan University, Joondalup, WA 6027, Australia; 4School of Public Health, Curtin University, Perth, WA 6845, Australia; gavin.f.pereira@curtin.edu.au; 5Neonatal Directorate, Perth Children’s Hospital, Hospital Avenue, Perth, WA 6005, Australia

**Keywords:** probiotics, *Bifidobacteria*, *Lactobacillus*, caesarean section, vaginal seeding, neonate, dysbiosis

## Abstract

The first few days following the birth are a vulnerable time for the neonate. Sick infants experience various interventions during their stay in the neonatal unit in order to stay alive and grow. Acquisition of gut microbes is critical for the short- and long-term health of the neonate. At a time when the gut microbiome is starting to take shape, crucial interventions directed at improving the growth, development and survival of the neonate impact its development. Events prior to and after the birth of the neonate, such as maternal conditions, antibiotic exposure, type of feeds, supplemental probiotics, and neonatal intensive care environment, contribute significantly to shaping the gut microbiome over the first few weeks and maintain its healthy balance crucial for long-term health. In this comprehensive review, we address common interventions the neonate is exposed to in its journey and their impact on gut microbiome, and discuss various interventions that minimize the dysbiosis of the gut.

## 1. Background

Various microorganisms comprising commensal bacteria, fungi and viruses colonize the human gastrointestinal tract, which are collectively referred to as the “gut microbiota” [1,2]. “Gut microbiome” refers to the collection of genomes from the entire gut micro-biota [2,3]. The gut microbiome develops over the host’s lifetime, with the early neonatal period being the most critical period for its development. It was long considered that the newborn baby’s gut is sterile at birth. Identification of bacterial products in the meconium prompted researchers to look for the exact timing of gut colonization [4]. Recent evidence indicates that the process of gut microbial development is initiated in utero and continues after birth [5]. During the process of birth, as the fetus descends into the birth canal, it comes in contact with the maternal perineal bacteria, thus initiating neonatal skin colonization and oropharyngeal colonization [6]. This lays the foundation for the development of gut microbiota. Factors conducive and non-conducive to the growth and multiplication of commensal bacteria would then contribute to further development of gut microbiota [7], especially in the early developmental phase from birth to 14 months of age [8].

*Bifidobacteria* and *Lactobacilli* form the major component of gut microbiota. The gut microbiota is considered healthy if a combination of diverse bacteria and commensal fungi are present in a state of balance. If the right balance is not maintained, it is referred to as dysbiosis [9]. The dysbiotic gut has been shown to be associated with various adverse outcomes such as necrotizing enterocolitis in the neonatal age and predisposition to allergy in early infancy and childhood, to name a few [9,10,11]. Even in the long term, dysbiosis is shown to be associated with metabolic dysfunctions and obesity [12]. Hence, researchers worldwide have identified strategies to promote the initiation and development of healthy gut microbiota in the neonatal age group and reduce the extent of dysbiosis [13,14]. Given its significance, a thorough understanding of factors that promote and inhibit gut microbial growth [7] in this early phase of life is crucial to identify interventions to attenuate dysbiosis, which, in turn, could improve clinical outcomes [13,15].

Over the last decades, improvements in care provision to the pregnant mother and fetus and early identification of risks have resulted in many interventions, thus reducing perinatal, neonatal, and maternal mortality and morbidity [16,17]. In recent years, improved understanding of the gut microbiome and its role in shaping the health journey of the infant has opened many treatment options [18,19]. Interventions are directed to support the development of gut microbiota resembling that of a vaginally born infant with minimal medical intervention, often considered the reference standard. While some interventions during labor, such as caesarean section (CS), antibiotic exposure prior to CS, and intrapartum antibiotic prophylaxis for the prevention of group B streptococcal sepsis, are now standard and largely non-modifiable, other factors, such as use of mother’s own milk, or donor milk, rational use of antibiotics, and oral supplementation with probiotics, could be modified to promote healthy microbial development.

Given the importance of establishing a healthy gut microbiome in a neonate, we aimed to comprehensively review the various factors that affect the early developmental pathways of gut microbiota in the neonatal age group (birth to 28 days of life).

Search strategy: A Medline search was carried out using the words “Gut Microbiome”, “probio*”, “neonate”, “preterm”, “infant”, “vaginal” and “caesarean section”. Human studies published in the English language with clinical trials or a randomized controlled trial design were selected. The search was carried out in other databases, such as the Cochrane library and Embase, and the results were cross-checked. Only relevant publications for this manuscript’s content were selected. A Medline search for existing systematic reviews was conducted on 20 November 2024 and repeated on 2 April 2025. MeSH words “microbiome” with the limit ‘neonates’ was used. A total of 56 relevant systematic reviews were identified in the initial search. Since this was a comprehensive clinical review and not a systematic review, we selected articles deemed relevant by the authors published in the last 10 years.

In the following sections, we discuss some of the common interventions and their im-plications on the development of gut microbiota in neonates (Figure 1). Important findings from recent systematic reviews on common neonatal interventions and gut microbiota are presented in Table 1.

## 2. Mode of Delivery and Gut Microbiota

It has been recognized that neonates born by CS have different compositions of gut microbiota compared to those born vaginally [27,28,29]. Neonates born by CS are also likely to experience longer-term health problems [30]. A study by Shao et al. reported that in infants born vaginally, the genera of Bifidobacterium in the gut were significantly higher, whereas those born by CS had a higher concentration of *Enterococcus*, *Staphylococcus*, *Streptococcus*, *Klebsiella*, *Enterobacter cloacae*, and *Clostridium perfringens* [31]. No difference in gut microbiota was observed in infants born by elective vs. emergency CS [31], thereby pointing towards a factor that is common for both elective and emergency CS. A comparison of infants born by CS with vaginally born infants exposed to intrapartum antibiotic prophylaxis identified many similarities between the two groups, pointing towards the likelihood of antibiotic exposure as the main event that shaped the gut microbiota [31]. During the neonatal period, babies born by CS were more likely to carry opportunistic pathogenic species in their gut compared to those born vaginally [32]. Yassour et al. reported that infants born by CS had a higher prevalence of Bacteroides species compared to vaginally born babies [33]. Infants born by CS have a higher likelihood of antibiotic exposure, and increased risk as well as duration of hospitalization, which, in turn, impact the composition of gut microbiota. Such changes in gut microbiota were observed until 1-2 years after birth, indicating the long-lasting impact of the mode of delivery in shaping the microbial architecture in infancy [32].

Although it is reasonable to assume that vaginally born infants are the reference population for gut microbial composition and development in neonates, one needs to be aware of the non-modifiable factors associated with vaginal births. Previous studies evaluating the gut microbiota in vaginally born neonates may not have excluded pregnancies with complications, such as maternal infections, prolonged rupture of membranes, antibiotic use and other medical conditions, that may adversely affect the microbial development after birth. Hence, healthy development of gut microbiota may not be equally distributed or sustained across all vaginal births.

## 3. Antibiotic Exposure in the Neonatal Age Group (0–28 Days)

Antibiotic administration to pregnant mothers during labor, also referred to as intrapartum antibiotic prophylaxis (IAP), is a common practice if mothers are colonized with group B *Streptococcus* (GBS) during pregnancy [34]. The American Academy of Pediatrics recommends administration of antibiotics to pregnant mother colonized with GBS identified in a lower vaginal swab performed at 36 weeks of gestation. IAP has resulted in decreased neonatal mortality and morbidity due to GBS infection in neonates [35]. Over recent years, research has identified that neonates exposed to IAP show lower bacterial diversity, a lower abundance of Actinobacteria, especially *Bifidobacteria*, and a larger abundance of Proteobacteria in their intestinal microbiota compared with infants not exposed to IAP [20,36]. Combellick et al. reported that the gut microbiota profile differed in home-born vs. hospital-born neonates. The authors compared the fecal microbiota of vaginally born neonates, 14 born at home and 21 born in the hospital. The hospital-born infants showed reduced diversity and an increased Lactobacilli count [37]. The authors identified that it was the interventions carried out in the hospital and hospital environment that determined the neonatal gut microbial development in the first 28 days of life [37]. In many countries, IAP is a non-modifiable risk factor, has been standard practice for years, and it is a key performance indicator for maternal care [38]. Hence, there is a need to identify strategies that minimize its unintended effects on neonates.

The neonatal nursery is designed to be a safe place for sick neonates who require monitoring and support. It is also a vulnerable period for the infant, and the risk of morbidity and mortality from infections and sepsis is high [39]. Improvement in care provision, especially to preterm infants, has resulted in improved survival and reduced mortality and morbidity from sepsis-related causes [40]. Improved understanding of relative immune insufficiency in preterm neonates, vulnerability of the premature infant to the rapid spread of infection, and promotion of quick use of antibiotics after birth in sick neonates has resulted in improved survival. Hence, over the years, the use of antibiotics in neonates has become standard practice. Administration of intravenous antibiotics within the first 2 h of birth was considered very important and the time to the first dose of antibiotics was considered a key performance indicator until recently [38,41]. However, recent reports have indicated that exposure to antibiotics in the first few days after birth induces dysbiosis of the gut microbiota [42]. A recent systematic review by Mulinge et al. reported that preterm infants exposed to antibiotics after birth showed reduced bacterial diversity, decreased or absence of Bifidobacteria species, a higher likelihood of presence of pathogenic bacteria such as Enterobacteriaceae, and a reduction in the alpha diversity of probiotic bacteria in the stools [21]. Alpha diversity is a measure of bacterial richness and bacterial variety upon stool testing, and it is a measure of dysbiosis. The above study reported that dysbiosis was sustained over the first few weeks after birth. Hence, one needs to be careful with the decision of starting antibiotics in neonates.

In the following sections, we present various strategies to improve healthy bacterial colonization in neonates and reduce dysbiosis.

## 4. Risk-Based Approach for Starting Antibiotics

A large number of neonates are started on intravenous antibiotics to treat presumed sepsis based on clinical risk factors. “Rule out sepsis” is one of the common categories of patients admitted to the neonatal unit. Since the awareness of negative effects of antibiotics on gut microbiota in vulnerable newborn infants has increased, a risk-based approach to determine the need for parenteral antibiotics in newborn infants immediately after birth has been trialed. It is hoped that this will reduce the large number of infants exposed to antibiotics. The advent of sepsis calculator derived from data collated from a large number of patients from the Vermont Oxford Network translates clinical risk into numerical risk [43]. Some of the clinical risk factors, such as maternal infection, fever, and GBS status, are used to arrive at the numerical risk, which is individualized for each patient. The presence of clinical symptoms of sepsis in the algorithm increases the sensitivity of such a diagnosis of likelihood of early sepsis. The initial concern with the sepsis calculator was that such strategies would increase the likelihood of missing cases of early neonatal sepsis. Clinical reports indicate that the use of the sepsis calculator is safe and does not increase the risk of missed diagnoses. Use of the sepsis calculator also resulted in a lower number of babies exposed to antibiotics [43], thus contributing to reducing dysbiosis in newborn infants.

Once antibiotics are initiated, it is common practice to continue them for 48 h until the blood culture results are available. Research shows that the longer the duration of antibiotic exposure, the worse the impact on gut microbiota [44]. In fact, a study by Rooney et al. reported that each additional day on antibiotic exposure in newborn infants was associated with a 16 % reduction in obligate anaerobic bacterial load in stools measured within a week of discontinuation of antibiotics [45]. Zwittinik et al. showed that >5 days of antibiotic exposure was associated with decreased abundance of *Bifidobacteria* [44]. Thus, one needs to be careful before starting antibiotics, and, once started, its discontinuation as early as possible is crucial to assist the development of the gut microbiota in neonates.

## 5. Skin-to-Skin Care

Skin-to-skin care, also referred to as Kangaroo care, is the practice of placing the newborn baby on the caregiver’s chest for a brief period on a regular basis. This is practiced in term and preterm infants who are clinically stable. Skin-to-skin contact has been shown to promote maternal and neonatal bonding, improve the rates of breastfeeding and maternal breast milk output, and lead to neonatal temperature regulation and stabilization of the heart rate [46].

Kangaroo care has also been shown to exert a beneficial effect on the development of the gut microbiota in the neonate. Govindarajan et al., in their pilot RCT, reported that kangaroo care decreased the presence of pathogenic bacteria such as E. coli and increased the relative abundance of Firmicutes, thus improving the microbiological environment [47]. In a secondary analysis of an RCT, where healthy full-term infants were offered one hour per day of skin-to-skin contact for five weeks, Eckerman et al. observed reduced microbiological volatility, thereby keeping the microbial balance steady [48]. Hendricks- Munoz et al. reported that skin-to-skin contact improved the oral microbiological composition of the neonate by promoting the growth of *Streptococcus*, while patients without skin-to-skin contact showed the presence of Corynebacterium and Pseudomonas [49]. In fact, Lamy Filho et al. reported that neonates whose skin was colonized with Methicillin-resistant Staphylococcus demonstrated better clearance of bacteria after regular kangaroo care [50]. Thus, skin-to-skin care is a safe practice that may improve gut microbial colonization in the neonate.

## 6. Probiotic and Prebiotic Supplementation

In the late 1990s to early 2000s, awareness of the presence and role of commensal bacteria in the neonatal gut grew. The presence of *Bifidobacteria* and *Lactobacilli* in higher quantities was observed in the gastrointestinal tract of preterm infants who were breastfed compared with formula-fed infants [51,52]. Promoting the growth of commensal probiotic bacteria in the gut by oral administration of live probiotic bacteria in premature infants was studied in many randomized controlled trials. The results indicated clinical benefits and were consistent across developed and developing countries. Clinical trials, systematic reviews, and meta-analyses reported a reduction in the risks of necrotizing enterocolitis, an inflammatory condition of the intestines observed in neonates [53]. This resulted in the introduction of oral probiotic bacteria containing a combination of strains of *Bifidobactria* and *Lactobacilli* as a routine in preterm infants born under the gestational age of 34 weeks [54]. Vievermanns et al. reported microbiological benefits and showed increased relative abundance of *Bifidobacteria* and *Lactobacilli* that were used as supplemented strains and a reduction in pathogenic strains, thus demonstrating the efficacy of probiotic supplementation in neonates [25].

Over the years, various centers have used different strains and combination of probiotic bacteria. The quantity of supplemental probiotic bacteria has also been revised to provide the best possible outcome for the patient. Different centers have opted for different probiotic bacteria across the world. In Western Australia, routine probiotics implementation in preterm infants born under the age of 34 weeks of gestation was initiated in 2011. *Bifidobacterium brevi* M 16, a single strain, was the product of choice until 2024, when it was changed to a triple strain consisting of Bifidobacterial species. Preterm infants ready to be fed orally are administered a once-daily dose of the probiotic supplement. Supplementation is continued until discharge from the hospital unless a contraindication for supplementation arises. Other states in Australia use a combination of Bifidobacterial species consisting of *Bifidobacterium bifidum*, *Bifidobacterium infantis* and *Lactobacillus* [55,56]. In Canada, a combination of strains of *Bifidobacteria* and *Lactobacillus* has been used [57]. A study by Alshaikh that summarized the probiotic administration and outcomes in Canadian neonatal network reported the use of multi-strain probiotics (FloraBABY, RenewLife) containing four *Bifidobacterium* strains (*Bifidobacterium reve*, *Bifidobacterium bifidum*, *Bifidobacterium longum*, and *Bifidobacterium infantis*) and *Lactobacillus rhamnosus*. The report also mentioned uniformity in the choice of probiotics across centers [58]. It appears that the rationale for choosing different combinations depended on its local availability, and the perceived advantages of one over the other [59]. Despite the use of different strains of probiotics, the intention of the treatment remains to modulate the gut microbiota in preterm infants. The short-term benefits of supplementation of probiotic bacteria have remained consistent, highlighting the ongoing benefits [60]. However, the effect of probiotic supplementation on long-term gut colonization has been questioned [61].The American Academy of Pediatrics opines that while probiotic supplementation results in a positive effect on the gut microbiome, its clinical benefits and adverse effects need to be carefully considered in centers that routinely use them [62].

He et al., in their recent systematic review and meta-analysis of five microbiome data sets from stool analysis published in 2024, reported that preterm infants who were treated with probiotic supplementation showed enrichment of *Acinetobacter*, *Bifidobacterium*, and *Lactobacillus* species and the depletion of the potentially pathogenic bacteria *Finegoldia*, *Veillonella*, and *Klebsiella* species. Gradually, *Bifidobacteria* became a dominant species even though preterm infants were exposed to the same doses of both *Bifidobacteria* and *Lactobacilli*. The clinical implications of this findings are uncertain [63]. Nevertheless, use of exogenous probiotic bacteria has been shown to promote beneficial bacterial colonization in infants born vaginally or by CS. Thus, probiotic supplementation could be a strategy to minimize the effects of dysbiosis in newborn infants.

Administration of probiotics in vulnerable newborn babies is not without risk. Preterm infants are vulnerable to immune modification and probiotic bacteria that are alive have the potential to cause sepsis in very preterm infants. Alshaikh et al., in a retrospective cohort study from Canada involving more than 18000 infants born <34 weeks of gestation, reported that a total of 27 preterm infants were diagnosed with probiotic sepsis [58]. Of the 27 infants, 20 were <1000 grams at birth and exogenously administered probiotic *Bifidobacterium brevii* [58]. The long-term effects of probiotics on immune functions in a neonate have not been well studied and practitioners should be aware of the rare but potential risk in vulnerable infants.

Prebiotic oligosaccharides (Pre OS) are non-soluble fibers that act as substrates for the multiplication of probiotic bacteria [64]. Pre OS, when ingested, escape the acidity of the stomach intact and reach the large intestine, where they act as substrates for probiotic bacteria. Galacto-oligosaccharides, fructo-oligosaccharides, and inulin are some of the Pre OS that have been trialed. A systematic review by Kebbe et al. that included 30 studies and 5290 patients reported that Pre-OS-enriched formula increased the count of *Bifidobacteria* but its effect on *Lactobailli* was inconsistent [24]. A combination of galacto-oligosaccharides and fructo-oligosaccharides in different concentrations has shown clinical benefits in newborn infants. Use of prebiotics seems to be most beneficial in combination with probiotics, and this is termed “synbiotics”. Trials have demonstrated an improved gut microbiota profile in infants who were administered Pre OS and probiotic-rich formula milk [65] and this can be a strategy to improve the gut microbiota in preterm and term infants.

## 7. Neonatal-Feeding Practices

The positive impact of human breast milk on the gut microbiota is well known. Breast milk contains prebiotic oligosaccharides and live probiotic bacteria that promote the healthy development of gut microbiota. Often, breastfed babies are considered the gold standard and act as reference standard for studying the gut microbiome. The microbial composition of breast milk has been shown to vary with the stage of lactation and postnatal age, with Streptococcus being most abundant in week 1 and *Bifidobacteria* and Lactobacilli being most abundant in week 4 [66]. Many clinical trials have reported the short-term and long-term benefits of breastfeeding in promoting the health of the newborn infant [67]. Inchingolo et al., in their systematic review, reported that infants who were breastfed showed a healthier, diverse microbial composition compared to formula-fed infants, who showed increased numbers of *Enterobacteriaceae* and *Clostridium difficile* [22]. At times, it is not possible to start feeding newborn infants due to sickness or surgical conditions, and, in such scenarios, trophic feeding, where a small amount of breast milk is administered into the baby orally or through the gastric tube on a regular basis, has shown enormous clinical benefits in reducing dysbiosis of the gut microbiota.

In recent years, donor milk (DM) banks across the world have made human milk available to preterm infants when the mother’s own milk is not available. The supply and administration of DM to vulnerable newborn infants is strictly controlled. It is expected that DM provides benefits similar to those of breast milk. DM is pooled from many donors and is pasteurized to inactivate pathogenic bacteria and has an unintended effect of neutralizing the probiotic bacteria present in breast milk. In a study comparing the microbiological profile of preterm infants fed with DM, mother’s milk, or formula milk, Anna Parra-Llorca et al. reported healthier development of gut microbiota in infants fed with DM or mother’s milk compared to formula milk [68]. Compared to formula milk, DM seems superior in promoting the growth of healthy bacteria in the gut [68]. Chen et al. reported higher alpha diversity in infants exposed to DM in the first few days and it seemed to decrease over time, prompting the authors to conclude that the use of DM reduced the alpha diversity compared with the use of mother’s own milk in the few weeks after birth [23]. Piñeiro-Ramos et al. reported that newborn infants fed with donor milk showed higher concentration of *Staphylococcus genus* and *Pasteurellaceae* [69]. In a systematic review, Cartagena et al. noted the benefits of DM on the gut microbiota and called for standardized and evidence-based guidelines [70], which are currently being developed. In the future, personalization of DM by adding some of mother’s own milk or even specific components of human milk to improve the quality of DM should be considered [71].

Use of calorie-dense fortifiers to promote the physical growth of preterm infants is a common practice in neonatal intensive care units. Added calories could be derived from cow’s milk or human milk. This practice of human milk fortification increases the protein and calorie content of milk used for feeding the newborn infant [72]. Such a practice also impacts the development of gut microbiota. Stinson et al. reported that the growth of *Firmicutes* and *Klebsiella* was higher in infants fed with human-milk-derived calories [71].

Iron supplementation in the form of ferrous sulphate is commonly prescribed in preterm infants [73]. The authors of a recent systematic review reported a 10.3% (95% confidence interval (CI): −15.0—5.55%) reduction in the *Bifidobacteria* population upon stool analysis in the iron group and a 2.96% reduction for the non-iron group. The studies were heterogenous, publication bias was not ruled out, and only one trial conducted in a neonatal population was included [73].

## 8. Vaginal Seeding

The gut microbiota composition in babies born by CS has been shown to be different from that of infants born by the vaginal route. In order to improve the gut microbiological composition in such infants, exposing them to vaginal fluid containing vaginal microbiota immediately after birth has been trailed [74]. Gauze is placed in the mother’s vagina for a prespecified length of time and is removed before the birth of the baby. The gauze is washed with saline and filtered to remove the debris and particles. With the resultant solution, the newborn baby’s mouth, palate, and face are smeared to inoculate with the vaginal microbiota. This technique is referred to as vaginal seeding and is performed only once after birth. In a pilot study, Dominguez-Bello et al. studied 11 infants born by CS and 7 infants born vaginally, exposed them to vaginal seeding, and studied the gut microbial profile [75]. The results indicated that infants born by CS exposed to vaginal seeding developed gut microbiota similar to vaginally born infants and it resulted in partial normalization of the gut microbiota [75]. The trials have shown mixed benefits with improving gut microbial composition. In a placebo-controlled RCT, Wilson et al. studied the effect of vaginal seeding on gut microbiota in CS-born babies at 24 h, 1 month, and 3 months by analyzing the stool for microbial DNA. Babies born by the vaginal route showed a higher concentration of Bacteroides compared to CS-born infants. In infants who were exposed to seeding at birth, the gut microbial composition was comparable to that of CS-born infants with and without seeding. The authors indicated that the mother’s vaginal microbes did not seem to colonize the neonatal gut [76] but the procedure was safe to perform [77].

A recent systematic review by Wang et al. reported on the modest changes observed in the gut microbiota after vaginal seeding and called for more clinical studies to understand the implications [78]. Further clinical research is required to understand the long-term positive and negative impacts of vaginal seeding on neonates. At this stage, vaginal seeding remains an experimental practice. The American College of Obstetrics and Gynecology guidelines call for caution before this practice is clinically implemented [79].

## 9. Neonatal Nursery Environment

The makeup of the neonatal nursery has changed over the years. In the past, neonatal intensive care units (NICUs) were open-plan, and single-room beds were limited to infants affected by infections that could spread from patient to patient by fomites. Over the years, neonatal nurseries have moved to single rooms as a matter of preference. This move was considered to reduce cross-contamination and hospital-acquired infections, and minimize the noise and disturbance experienced by the fragile infant. Researchers have reported on the gut microbiota in infants exposed to various environmental triggers in the neonatal nursery, including those nursed in single rooms vs. open-plan environments [26]. In a cohort study, Brooks et al. studied the microbial signature from the room environment by analyzing 2882 samples collected from the neonatal unit occupied by very-low-birth-weight infants. The authors observed differences in gut microbiota in each room and reported that each room seemed to possess a unique microbial signature [80]. Equipment and the NICU environment seemed to influence the acquisition of bacteria identified on swabs collected from the work surfaces in the NICU, and that closely reflected the gut and skin microbiota in premature infants [26]. Thus, each NICU is unique and has its own microbial signature. Research indicates that the neonatal unit environment determined the clostridial colonization in preterm infants’ gastrointestinal tracts and that the colonization of the surfaces is also influenced by the unit’s antibiotic use [81,82]. de Goffau et al. reported that the neonatal humidifier cots harbored specific bacteria that colonized in colder areas of the humidifiers [83]. In fact, the PIPS trial—a large RCT that studied the effect of Bifidobacterium supplementation in preterm infants—demonstrated a high cross-contamination rate of 49% observed in the control population not exposed to probiotic supplementation [84]. Thus, the NICU environment plays an important role in the process of gut microbial development in newborn infants.

Other important factors such as maternal diet and maternal nutrition during late pregnancy have been shown to induce changes in the neonatal gut microbiota. High-fat diets, the Mediterranean diet, and vegetarian diets have been researched and correlations with increased or decreased levels of certain types of bacteria in the neonatal gut have been reported [7,85]. Neonate-specific factors, such as the use of thickeners to treat gastro-esophageal re-flux [86], presence of siblings [87], and parental home environment factors, such as the presence of pets [88], have also been shown to impact the development of the gut microbiota. All these factors need to be considered and modulated to ensure optimal colonization of gut with healthy commensals [89].

## 10. Conclusions

Implications on clinical practice: In summary, various perinatal and postnatal factors contribute towards the development of the gut microbiota in newborn infants and have the potential to cause dysbiosis. Clinical interventions that help initiate healthier gut microbial development and reduce dysbiosis in preterm newborn infants have become important in care provision in neonatal intensive care units. Further well-powered RCTs are necessary to study the long-term effects of such interventions on gut microbial dysbiosis.

## Figures and Tables

**Figure 1 microorganisms-13-01772-f001:**
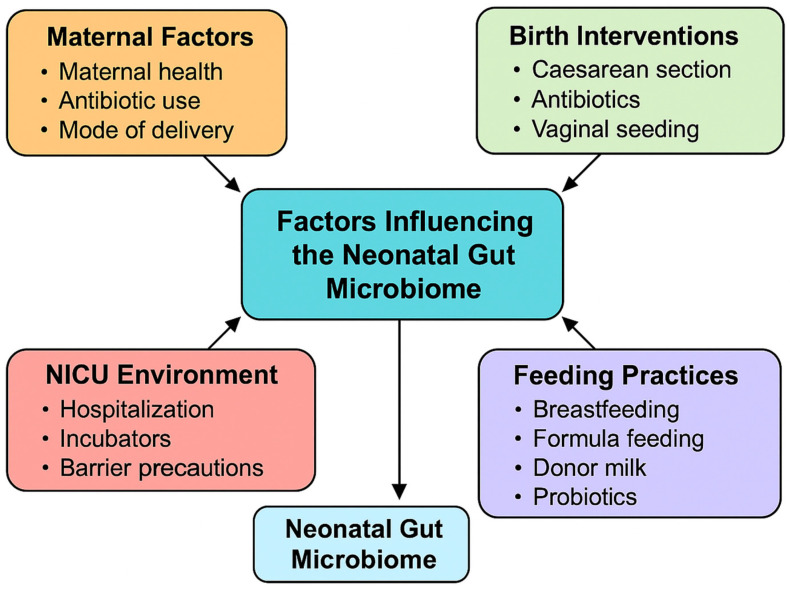
Factors affecting the development of neonatal gut microbiome.

**Table 1 microorganisms-13-01772-t001:** Neonatal interventions and their effect on gut microbiota.

Neonatal Interventions	Key Systematic Review [Reference] Author Year	Results and Comments
Delivery by caesarean section	Pivrncova et al., 2024 [20] No of studies 14	Infants born by caesarean section showed depletion of *Bacteroides.* This result was observed despite breastfeeding in the first 3 months. In total, 48% of breastfed infants born by vaginal delivery vs. 19% infants born by caesarean section showed predominance of *Bifidobacteria* in the first 2 weeks, which persisted up to 3 months.
Exposure to antibiotics in neonatal age	Mulinge et al., 2023 [21] No of studies 21	Preterm infants treated with cephalosporin antibiotics showed a reduced Shannon index, a measure of alpha diversity, reduced evenness of bacterial species identified in stool specimen, decreased abundance of Bifidobacteria species, increased pathogenic bacteria such as Enterobacteriaceae and Bacteroides, and an increase in *Staphylococcus* spp., *Streptococcus* spp., *Serratia* spp., and *Parabacteroides* spp. compared with controls.
Breastfeeding infants and formula feeding	Inchingolo et al., 2024 [22] No of studies 13	Breastfed infants showed higher levels of *Bifidobacterium* and *Lactobacillus*, while formula-fed infants had a higher prevalence of *Clostridium* and *Enterobacteriaceae* considered to be potentially pathogenic.
Donor human milk	Chen et al., 2024 [23] No of studies 12	Donor-milk-fed infants showed reduced diversity of bacteria. The Shannon index and Gini–Simpson index was used as measures of alpha diversity of stool samples from birth to day 60 of life. Donor-milk-fed infants showed higher abundances of *Staphylococcaceae* and *Clostridiaceae* and lower abundances of *Bacteroidetes* and *Bifidobacterium* compared to breastfed infants. At one month of life, concentrations of fecal metabolite such as propionate were higher and those of acetate were lower in the donor milk group.
Formula supplemented with Prebiotics	Kebbe et al., 2025 [24] Number of studies 30	Use of prebiotic oligosaccharide in infant formula compared with standard formula showed increased Bifidobacterium counts (k = 7 [MD: 0.49; 95% CI, 0.27–0.71]; I2 = 13% and decreased fecal pH. Use of fructo oligosaccharides showed variable results on the counts of *Bifidobacteria*, with mild increase in some trials and no difference observed in other trials. Meta-analysis showed no difference in the mean counts of *Lactobacilli*, *bifidobacterium* species in prebiotic vs. human milk-fed infants.
Probiotic supplementatio on	Vievermanns et al., 2024 [25] No of studies 29	Probiotic supplementation with *Bifidobacteria* and *Lactobacilli* led to increased relative abundance of probiotic strains used for supplementation. *Clostridium*, *Streptococcus*, *Klebsiella* and *Escherichia genera* were decreased in abundance in probiotic-exposed infants.
Neonatal intensive care environment	Hartz et al., 2015 [26] No of studies 11	In an intensive care environment, including ventilation, etc., tubing was colonized with *Streptococcus*, *Staphylococcus*, *Neisseria*, *and Enterobacteriaceae*, which was reflected in the gut microbiome profile of infants. Incubators were likely to be colonized with staphylococcus. This reflected in the gut microbiome of neonates and showed increased *Clostridia* and *Escherichia*, and reduced *Bifidobacteria.*

## Data Availability

No new data were created or analyzed in this study. Data sharing is not applicable to this article.

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
