# Peer review of "Management of Neonates in the Special Care Nursery and Its Impact on the Developing Gut Microbiota: A Comprehensive Clinical Review"

_microorganisms, 2025, doi:10.3390/microorganisms13081772_

Round 1
Reviewer 1 Report
Comments and Suggestions for Authors
The manuscript presents a highly relevant topic for guiding health professionals in the treatment of newborns. Some suggestions should be considered by the authors, especially considering the specificity of strains (which should be declared in the text) and the vulnerability of the babies (making very clear and precise information mandatory).
Comments on specific excerpts follow.
Line 176 "Streptococcus, while patients without skin-to-Staphylococcus demonstrated better clearance of bacteria after regular kangaroo care [40]. Please use italic for genus and species.
Lines 109-192 "This resulted in the introduction of oral probiotic bacteria containing a combination of strains of Bifidobacteria and Lactoba-
cilli as a routine in preterm infants born under the gestational age of 34 weeks[44]". It is convenient to talk about the specific strains and the risks associated with probiotic supplementation in general.
Lines 193-194 "Vievermanns et al reported microbiological benefits and showed increased relative abundance of supplemented strains and reduction in pathogenic strains thus demonstrating efficacy of probiotic supplementation in neonates [45]". Please include strains and risks.
Line 205 "In Canada, a combination of strains of Bifidobacteria and 205
Lactobacillus has been used [47]." Please, it is very important to include strains, or at least the trademarks of the product(s) for more accurate information and to better guide health professionals.
Line 211 "He et al" - include the year of publication.
Line 226 "newborn infants So far, the best use of". Include a period after newborn infacts.
Lines 245-246 "Sine DM is pooled from many donors, it is pasteurized to remove pathogenic bacteria. This may eliminate" Change "remove" and "eliminate" to inativate.
Line 256 Firmicutes - not italic
Line 256 "with human milk-derived calories [58]". What do you mean? Could you explain better?
Line 268 "inoculate the vaginal microbiota". I suggest "To inoculate *with*"
Line 299 "NICU environment. What is NICU?
Line 307 "bifidobacterium" Italic and capital
Line 320 "commensals[74]" Include period
Author Response
Reviewer 1.
The manuscript presents a highly relevant topic for guiding health professionals in the treatment of newborns. Some suggestions should be considered by the authors, especially considering the specificity of strains (which should be declared in the text) and the vulnerability of the babies (making very clear and precise information mandatory).
Response: Thank you for the comments. The manuscript has been amended extensively to reflect the suggestions. Wherever required the strain of bacteria, genus and species have been included.
Comments on specific excerpts follow.
Line 176 "Streptococcus, while patients without skin-to-Staphylococcus demonstrated better clearance of bacteria after regular kangaroo care [40]. Please use italic for genus and species.
Response: Edited
Lines 109-192 "This resulted in the introduction of oral probiotic bacteria containing a combination of strains of Bifidobacteria and Lactobacilli as a routine in preterm infants born under the gestational age of 34 weeks [44]". It is convenient to talk about the specific strains and the risks associated with probiotic supplementation in general.
Response: A new paragraph highlighting the risks and probiotic related sepsis has been added from line 286- 298 under the probiotic section. Detais of specific strains of probiotics used has been discussed form line 259 onwards.
Lines 193-194 "Vievermanns et al reported microbiological benefits and showed increased relative abundance of supplemented strains and reduction in pathogenic strains thus demonstrating efficacy of probiotic supplementation in neonates [45]". Please include strains and risks.
Response: The sentence has now been amended to include the strains.
Line 205 "In Canada, a combination of strains of Bifidobacteria and 205
Lactobacillus has been used [47]." Please, it is very important to include strains, or at least the trademarks of the product(s) for more accurate information and to better guide health professionals.
Response: Thank you.”. Details of the strain used in Canada is now included from line 271-277.
Line 211 "He et al" - include the year of publication.
Response: Year of publication 2024 included.
Line 226 "newborn infants So far, the best use of". Include a period after newborn infants.
Response. Edited.
Lines 245-246 "Sine DM is pooled from many donors, it is pasteurized to remove pathogenic bacteria. This may eliminate" Change "remove" and "eliminate" to inativate.
Response: Edited as suggested.
Line 256 Firmicutes - not italic
Response; Edited as suggested.
Line 256 "with human milk-derived calories [58]". What do you mean? Could you explain better?
Response: Thank you. Line 355- 358 have now been added to explain the human milk fortification to make it easier to understand and provide context.
Line 268 "inoculate the vaginal microbiota". I suggest "To inoculate *with*"
Response: changed as suggested.
Line 299 "NICU environment. What is NICU?
Response- The short form is now expanded.
Line 307 "bifidobacterium" Italic and capital
Response: changed as suggested.
Line 320 "commensals[74]" Include period
Response: changed as suggested.

Reviewer 2 Report
Comments and Suggestions for Authors
Ravisha Srinivasjois and co-authors submitted a manuscript to Microorganisms which reviews the gut microbiome of neonates.
The topic of the manuscript is relevant and corresponds to the venue of the journal. However, the structure is not good enough to recommend this manuscript for publication in its current state.
1) Review articles should be interesting for different groups of readers, therefore they should contain non-trivial data - graphs, diagrams, tables, not the plain text. In the current state they are missing in the manuscript. The addition of graphs and tables, in which the analysis of literary data is carried out, significantly increases the value of the review article and the number of the citations.
2) The review contains 19 subsections, 17 of which are related to the topic of the article (and also Background and Conclusions subchapters). Unfortunately, most of these subsections are one paragraph long. Do the authors have nothing more to add? Are the authors really experts on the topic? The reader will have reasonable doubts that this review contains all relevant information on the topic. I think the number of subsections should be reduced to 4-5 and the amount of information should be increased MULTIPLE, to at least 3-4 paragraphs on each topic. Currently, the text is about 6 pages, it is quite possible to increase it to 12 (not counting the first page, References, etc.).
In its current form, the article cannot be recommended for publication in Microorganisms.
Comments on the Quality of English LanguageProfessional scholar English editing will make the manuscript more useful to the readers
Author Response
Reviewer 2:
Ravisha Srinivasjois and co-authors submitted a manuscript to Microorganisms which reviews the gut microbiome of neonates.
The topic of the manuscript is relevant and corresponds to the venue of the journal. However, the structure is not good enough to recommend this manuscript for publication in its current state.
Response: Thank you for the comment. We hope that with the improvements in edits and flow, the manuscript quality has improved.
1) Review articles should be interesting for different groups of readers, therefore they should contain non-trivial data - graphs, diagrams, tables, not the plain text. In the current state they are missing in the manuscript. The addition of graphs and tables, in which the analysis of literary data is carried out, significantly increases the value of the review article and the number of the citations.
Response:Thank you. The manuscript has now been expanded to 4338 words. An illustration has also been added to create an impact. Hope this is adequate.
2) The review contains 19 subsections, 17 of which are related to the topic of the article (and also Background and Conclusions subchapters). Unfortunately, most of these subsections are one paragraph long. Do the authors have nothing more to add? Are the authors really experts on the topic? The reader will have reasonable doubts that this review contains all relevant information on the topic. I think the number of subsections should be reduced to 4-5 and the amount of information should be increased MULTIPLE, to at least 3-4 paragraphs on each topic. Currently, the text is about 6 pages, it is quite possible to increase it to 12 (not counting the first page, References, etc.).
In its current form, the article cannot be recommended for publication in Microorganisms.
Response: Thank you. The manuscript is now expanded, reformatted, edited, multiple subsections removed but major subsections still retained. We hope that you will find this editing adequate to publish this comprehensive clinically relevant review in your esteemed journal.

Reviewer 3 Report
Comments and Suggestions for Authors
The paper seems to be not directed to readers of the Microorganisms but rather to neonatologists. It is clearly declared in its summary: “In this comprehensive
review, we identify common interventions the neonate is exposed to in their journey, their impact on gut microbiome and discuss various interventions that minimise the dysbiosisof the gut”. Indeed, an information on neonate gut microbiota is rather scanty in the paper. Morever, a selection of the articles on neonate gut microbiota looks incomplete since the reviewing methodology was based only on Medline, and selecting 56 papers, the authors discussed only „some of the common interventions”. The paper should be directed to another MDPI journal with more clinical orientation.
Author Response
Dear Editors,
Thank you for the opportunity to submit our revised manuscript with reviewer comments addressed below.
Reviewer comments:
Reviewer 3:
The paper seems to be not directed to readers of the Microorganisms but rather to neonatologists. It is clearly declared in its summary: “In this comprehensive review, we identify common interventions the neonate is exposed to in their journey, their impact on gut microbiome and discuss various interventions that minimise the dysbiosisof the gut”. Indeed, an information on neonate gut microbiota is rather scanty in the paper. Morever, a selection of the articles on neonate gut microbiota looks incomplete since the reviewing methodology was based only on Medline, and selecting 56 papers, the authors discussed only some of the common interventions”. The paper should be directed to another MDPI journal with more clinical orientation.
Response: Thank you.
The manuscript is now edited. The background section has been expanded to provide relevant information about gut microbiome in neonates. We agree that the selection of articles is incomplete since this comprehensive review is more directed at practice rather than finding evidence/ summarising evidence using systematic review methodology.
We are happy for the journal to suggest any alternative MDPI journal that the editors think is appropriate.

Round 2
Reviewer 2 Report
Comments and Suggestions for Authors
The reviewer suggest that one or several figures and one or several tables should be added to the manuscript. This will make the contents more useful to the potential reader and will increase the potential number of citations.
In its current state, most of the information provided in the manuscript can be obtained from the abstracts of the references.
Compiling the data in tables and/or figures significantly increases the value of the article to the reader.
Author Response
Comment: The reviewer suggest that one or several figures and one or several tables should be added to the manuscript. This will make the contents more useful to the potential reader and will increase the potential number of citations.
In its current state, most of the information provided in the manuscript can be obtained from the abstracts of the references.
Compiling the data in tables and/or figures significantly increases the value of the article to the reader.
Response: Thank you. We have now added a new table that summarizes the neonatal interventions and the effect on gut microbiome. We hope the reviewers find it useful.
|
Neonatal Interventions |
Key systematic review Author year |
Results and comments |
|
Delivery by caesarean section
|
Pivrncova et al 2022 No of studies 14 |
Infants born by caesarean section showed depletion of Bacteroides. This result was observed despite breast feeding in the first 3 months. 48% of breastfed infants born by vaginal delivery vs 19% infants born by caesarean section showed predominance of bifidobacteria in the first 2 weeks which persisted up to 3 months. |
|
Exposure to antibiotics in neonatal age |
Mulinge et al 2023 No of studies 21 |
Infants exposed to antibiotics showed reduced Shannon index, a measure of alfa diversity, reduced evenness of bacterial species identified in stool specimen, decreased abundance of Bifidobacteria species, increased pathogenic bacteria such as Enterobacteriaceae, and Bacteroides, increase in Staphylococcus spp., Streptococcus spp., Serratia spp., and Parabacteroides spp. in preterm infants treated with cephalosporin antibiotics compared with controls. |
|
Breastfeeding Infants and formula feeding |
Inchingolo et al 2024 No of studies-13 |
Breastfed infants showed higher levels of Bifidobacterium and Lactobacillus while formula fed infants had a higher prevalence of Clostridium and Enterobacteriaceae considered to be potentially pathogenic. |
|
Donor human milk |
Chen et al 2024. No of studies-12 |
Donor milk fed infants showed reduced diversity of bacteria. Shannon index and Gini-Simpson index were used as measures of alpha diversity of stool samples form birth to day 60 of life. Donor milk fed infants showed higher abundances of Staphylococcaceae and Clostridiaceae, and lower abundances of Bacteroidetes and Bifidobacterium compared to breast fed infants. At one month of life, concentrations of fecal metabolite such as propionate were higher and acetate were lower in the donor milk group. |
|
Formula supplemented with Prebiotics |
Kebbe et al 2025 Number of studies-30 |
Use of prebiotic oligosaccharide in infant formula compared with standard formula showed increased Bifidobacterium counts (k = 7 [MD: 0.49; 95% CI, 0.27-0.71]; I2 = 13% and decreased fecal pH . Use of fructo oligosaccharides showed variable results on the counts of bifidobacteria with mild increase in some trials and no difference observed in other trials. Metaanalysis showed no difference in the mean counts of lactobacilli, bifidobacterium species in prebiotic vs human milk fed infants. |
|
Probiotic supplementatio on
|
Vievermanns et al 2024 No of studies-29 |
probiotic supplementation with bifidobacteria and lactobacilli led to increased relative abundance of probiotic strains used for supplementation. Clostridium, Streptococcus, Klebsiella and Escherichia genera were decreased in abundance in probiotic exposed infants. |
|
Neonatal intensive care environment |
Hartz et al 2015 No of studies 11 |
Intensive care environment including ventilation etc tubing was colonized with Streptococcus, Staphylococcus, Neisseria, and Enterobacteriaceae which reflected in gut microbiome profile of infants. Incubators were likely to be colonised with staphylococcus. This reflected in the gut microbiome of neonates and showed increased Clostridia, Escherichia,and reduced Bifidobacteria. |

Reviewer 3 Report
Comments and Suggestions for Authors
The paper quality is now improved after making changes proposed by the reviewer.
Author Response
Reviewer 3 comments:
The paper quality is now improved after making changes proposed by the reviewer.
Response: Thank you. We have added a table and a figure for easy reading.
We hope that the edited version is appealing enough.